# The Influence of Food Waste Rearing Substrates on Black Soldier Fly Larvae Protein Composition: A Systematic Review

**DOI:** 10.3390/insects12070608

**Published:** 2021-07-04

**Authors:** Indee Hopkins, Lisa P. Newman, Harsharn Gill, Jessica Danaher

**Affiliations:** School of Science, STEM College, RMIT University, Melbourne 3083, Australia; s3744095@student.rmit.edu.au (I.H.); lisa.newman@rmit.edu.au (L.P.N.); harsharn.gill@rmit.edu.au (H.G.)

**Keywords:** alternative protein, amino acid, Black Soldier fly, food waste, insect protein, macronutrients

## Abstract

**Simple Summary:**

The Black Soldier Fly (BSF) is a viable option for countering the environment detriments caused by food waste and can provide a sustainable protein source to feed the growing global population. This systematic literature review investigated the impacts of various foodstuffs and food waste rearing substrates on the protein and amino acid composition of BSF larvae. From the 23 articles included, BSF larvae fed ‘Fish waste Sardinella aurita’ for two days produced the highest total protein content at 78.8% and rearing substrates ‘Fruit and vegetables’ reported the lowest protein content at 12.9% of the BSF total mass. However, variation in rearing and analytical methodologies between each study potential undermines the extent to which the rearing substrates may have influenced the overall protein content of BSF larvae, their application in nutrition is still in its infancy.

**Abstract:**

The Black Soldier Fly (BSF) offers the potential to address two global challenges; the environmental detriments of food waste and the rising demand for protein. Food waste digested by BSF larvae can be converted into biomass, which may then be utilized for the development of value-added products including new food sources for human and animal consumption. A systematic literature search was conducted to identify studies investigating the influence of food waste rearing substrates on BSF larvae protein composition. Of 1712 articles identified, 23 articles were selected for inclusion. Based on the results of this review, BSF larvae reared on ‘Fish waste *Sardinella aurita*’ for two days reported the highest total protein content at 78.8% and BSF larvae reared on various formulations of ‘Fruit and vegetable’ reported the lowest protein content at 12.9%. This review is the first to examine the influence of food waste on the protein composition of BSF larvae. Major differences in larval rearing conditions and methods utilized to perform nutritional analyses, potentially influenced the reported protein composition of the BSF larvae. While this review has highlighted the role BSF larvae in food waste management and alternative protein development, their application in nutrition is still in its infancy.

## 1. Introduction

With the predicted expansion of the global population expected to reach 8.5 billion by 2030 and 9.7 billion by 2050 [1], we have a vital need to develop and provide a safe and sustainable food system. With an annual increase of 83 million people worldwide, it is estimated that a 70% increase in food production will be required to meet demand, resulting in increased competition for arable land and natural resources such as energy and water [2]. However, our agricultural sectors’ ability for growth, particularly in the production of sufficient and quality protein from traditional sources, is constrained by a deficiency of these key resources and presents as a serious challenge. In addition to our current food production system having been deemed as unsustainable from a growth perspective, it is also linked to adverse environmental implications, such as greenhouse gas emissions (GHG) and soil depletion [2].

A simultaneous global issue, despite being paradoxical to our food production and sector growth problem, is that much of the food produced is wasted. Whilst quantification of the extent of the problem is difficult due to lack of consistency in definitions and evaluation methods, it is estimated that as much as one-third of food produced for human consumption is wasted globally each year [3,4]. Food waste negatively impacts the environment in several ways including the net loss of finite resources such as land, water and fuel consumed during food production and distribution. In addition, foodstuffs discarded into landfill are also a contributor to GHG emissions (namely methane), making food waste a growing contributor to climate change [5,6]. It is estimated that food waste is contributing 4.4 gigatons of carbon dioxide (CO_2_) emissions into the atmosphere annually [6]. If put into context against national rankings, food waste would be the third-highest contributor of total GHG emissions after that of the United States and China [6]. At present, alternative treatment methods for food waste include incineration, fodder, anaerobic digestion and aerobic composting [7]. However, these methods are either not without their own environmental concerns or unable to be used in isolation to satisfy environmental needs in the long-term.

Another viable alternative to food waste treatment is Conversion of Organic Refuse by Saprophagous (CORS) technologies which use decomposer insects such as the Diptera *Hermetia illucens* L., also known as the Black Soldier Fly (BSF) larvae to manage organic waste. Deteriorated fruits and vegetables, municipal waste, crop waste, and industrial food-processing waste can be quickly and effectively digested by the BSF larvae [8,9,10,11]. This process of bioconversion diverts organic food waste from landfill to the production of biomass which can later be utilised for the development of value-added products. The BSF larvae have shown promising results in the production of biodiesel [12], fish feed [13] and have the potential to be used as an alternative source of protein for livestock [14]. The farming of mini livestock (i.e., insects) offers many benefits including high feed conversion ratios, and lower resource inputs in terms of energy, land and water requirements when compared to the farming requirements of traditional livestock (i.e., cattle) [15]. BSF larvae are also safe for human consumption [16], thus offering an opportunity to produce a new sustainable protein source for the growing global population.

Due to the variable nature of the nutritional composition of BSF larvae, the effect of various rearing substrates on BSF development needs to be investigated in further depth. This information can assist in the establishment of efficient growing and production practices in the future. To date, research on the nutritional protein value of BSF larvae fed food waste as the primary rearing substrate has not been synthesised. Thus, the purpose of this systematic literature review is to synthesise and investigate the influence that rearing substrates comprising of foodstuffs or food waste products have on the nutrient composition of BSF larvae.

## 2. Materials and Methods

### 2.1. Literature Search Strategy

The current review was performed in adherence to PRISMA-P (Preferred Reporting Items for Systematic review and Meta-Analysis Protocols) guidelines [17]. Scopus, Food Science and Technology Abstracts (FSTA), Web of Science, PubMed, Scifinder and ScienceDirect databases were used to search for articles investigating food waste and the nutritional composition of BSF larvae. The search was limited to articles published between 1 January 2000 and 30 October 2020.

Search terms were pilot tested before commencing the final search to ensure appropriate articles were identified. The final search included keywords searched within three categories (using ‘OR’) and then combined (using ‘AND’). One category searched for articles reporting data on the insect of interest (i.e., Black Soldier Fly, or *Hermetia illucens*). The second category searched for articles reporting on rearing substrates (i.e., food waste, or diets). The third category searched for articles reporting on nutritional outcomes (i.e., protein, amino acids).

### 2.2. Inclusion and Exclusion Criteria

To be included in the review, articles were required to meet the following criteria: (i) be published as an original research study, with full-text availability in English; (ii) reporting on BSF in the larval or pre-pupae life stage; (iii) including a by-product of food production, foodstuffs or food waste as the rearing substrate for BSF larvae; (iv) reporting an assessment of protein composition of the BSF larvae.

The exclusion criteria included: (i) opinion articles, reviews, narrative reviews and concept papers; (ii) abstract and conference proceedings where a full-text published article could not be obtained; (iii) studies published in a non-English language; (iv) data previously reported elsewhere; (v) studies rearing BSF larvae on alternative waste products not safe for human consumption (i.e., manure); (vi) studies rearing BSF larvae on foodstuffs or food waste products with the inclusion of microbial assistance (i.e., fermentation).

### 2.3. Study Selection

Articles detected using the search strategy were collated using EndNote. Two reviewers (IH and JD) screened all articles based on titles and abstracts initially, and then by a full-text review. Articles that did not meet the eligibility criteria were excluded. For any articles where it was unclear whether the eligibility criteria were met, full-text articles were obtained, screened and resolved by discussion between three reviewers (IH, LPN and JD) until a consensus was reached.

### 2.4. Data Extraction and Synthesis 

The data extraction method was pilot tested using an article retrieved whilst piloting search terms and refined accordingly. Data from the eligible articles were extracted using a Microsoft Excel spreadsheet, with the following study attributes recorded: year of publication, rearing substrates nutritional and physical characteristics, abiotic factors (including temperature, humidity, and light availability), BSF larvae rearing duration study methodology, statistical analysis, key results and author’s conclusions. Data extraction was performed by one reviewer (IH) and verified by secondary reviewers (LPN and JD). In cases of disagreement, a discussion was held until consensus was reached. Extracted data were unable to be combined in a meta-analysis as the studies were not sufficiently homogenous in terms of design and comparator.

### 2.5. Risk of Bias Assessment

Risk of bias assessment of all included articles was independently undertaken by two reviewers (IH and JD) using the SYRCLE risk of bias tool [18]. The assessment tool considered (i) whether the allocation sequence was adequately generated and applied; (ii) were the groups similar at baseline; (iii) was allocation adequately concealed; (iv) were the animals randomly housed during the experiment; (v) were caregivers and investigators blinded from the knowledge of which intervention each group received; (vi) were larvae selected at random for outcome assessment; (vii) was the outcome assessor blinded; (viii) was incomplete outcome data adequately addressed; (ix) was the study free of selective outcome reporting; (x) was the study apparently free of other issues that could result in high risk of bias. A ‘yes’ response indicated a low risk of bias, a ‘no’ response indicated a high risk of bias, and an ‘unclear’ response indicated that insufficient details had been reported to allow for a valid assessment of the risk of bias [18]. Any discrepancies in the risk of bias assessment were reviewed by a third reviewer (LPN) and resolved by discussion and consensus between the three reviewers.

## 3. Results

### 3.1. Search Results

A total of 1712 records were identified through database searches (n = 1051 duplicates), of which 98 were accepted for full-text review after screening by titles and abstracts (Figure 1). Of these, 76 articles were excluded due to the following reasons: BSF larvae was not the primary species investigated (n = 2), rearing substrate did not meet inclusion criteria (n = 17), rearing substrate included added supplementation with microbial assistance or fermentation (n = 2), measurable outcome did not include a relevant nutritional assessment of BSF larvae (n = 33), full-text was not available or was a conference proceeding (n = 10), article was not published in a scientific journal (n = 3) or was not a primary research study (i.e., review article, concept paper or patent application) (n = 9).

### 3.2. Rearing Substrates of Black Soldier Fly Larvae

Of the 23 articles included in the current review, 16 articles reported on rearing substrates that contained grain-based ingredients [19,20,21,22,23,24,25,26,27,28,29,30,31,32,33,34,35]. Fifteen articles reported on rearing substrates that contained fruit and vegetable ingredients [19,20,21,23,25,29,30,31,32,33,35,36,37,38,39]. Six articles reported on rearing substrates that contained animal-based ingredients [24,25,28,39,40,41], four articles reported on rearing substrates that contained a generic food or kitchen waste description, with no further details regarding included ingredients [24,38,39,41], and one article reported on a rearing substrate that contained seaweed as an ingredient [26].

The rearing duration of the BSF larvae in the experimental trials varied with a reported range between one day [40] and 52 days [37]. Details on larvae feeding frequency were provided by 19 articles, as shown in Table 1 [19,20,21,22,24,25,26,27,28,29,31,32,33,34,36,37,38,39,40]. Of these the feeding frequency ranged from a singular feed at the beginning of the experiment [19,35], to a set feeding schedule throughout the experimental trial including daily [26,34,36], weekly [32], and specific days [22,23,25,27,32,33], and ad libitum feeding approaches [21,29,30,37,39,40]. The feed ration provided to BSF larvae was reported by 15 articles [19,20,22,23,24,25,26,27,28,29,30,33,36,38,39] and varied considerably from 12.2 mg per larvae [23] to 1,530 mg per larvae [24].

### 3.3. Rearing Abiotic Conditions of Black Soldier Fly Larvae

Abiotic conditions of the 23 articles included in the current review are shown in Table 1. Of these, ten articles included details regarding duration of light and dark exposure hours of BSF larvae [19,20,21,25,26,27,29,30,34,36,37]. Five articles reporting a duration of 12 h of light and 12 h of darkness [20,21,27,30,37], two articles reporting 16 h of light and eight hours of darkness [19,36], one article reporting 24 h of light [29], one article reporting 24 h darkness [25] and one article reporting 14 h of light and ten hours of darkness [34]. Twenty articles included in this review provided data on the relative humidity of the BSF larvae rearing environment [19,20,21,22,23,24,25,26,27,28,29,30,31,32,33,34,35,36,37,38,39,40], with ranges varying from 40.0% relative humidity [27] to 75.6% relative humidity [23]. 

Twenty-two articles included information regarding the temperature of the rearing environment [19,20,21,22,23,24,25,26,27,28,29,30,31,32,33,34,35,36,37,38,39,40], with a range of 24.5 °C [27] to 32.5 °C [31]. The age of larvae at the beginning of the experimental trials was reported in 21 articles [19,20,21,22,23,24,25,26,27,28,29,31,32,33,34,35,36,37,38,39,40], with BSF larvae ages ranging from eggs which were directly inoculated onto the rearing substrate [22,36] to BSF larvae aged 14 days before being introduced to the substrate [25]. The rearing duration of BSF larvae was reported in 20 articles [19,20,21,23,24,25,26,27,28,29,30,32,33,34,35,36,37,38,39,40], with rearing duration ranging from eight days [26,29] to 52 days [37].

### 3.4. Macronutrient Composition of Rearing Substrates

Twenty of the 23 articles included in this review provided details on the total protein composition of the rearing substrate provided to the BSF larvae (Table 2) [19,20,21,22,23,24,25,26,27,28,29,30,31,32,33,34,35,36,37,38,39,40,41]. Eleven articles reported details of the rearing substrate total lipid content [19,24,25,26,27,28,29,30,33,34,37,39]. Thirteen articles additionally reported details of the rearing substrate total carbohydrate content [19,20,21,22,23,24,25,26,27,28,29,30,33,34,37,39].

There were substantial differences in the macronutrient composition of the rearing substrate provided, with an average total composition of 17.5% total protein, 7.2% total lipid and 27.3% total carbohydrate, when reported on a dry matter basis. Rearing substrate ‘Apple’ showed the lowest total protein content of 0.4% of dry matter [19]. The highest total protein content of a rearing substrate was reported for ‘Fish waste *S. aurita*’, with 72.7% protein dry matter [40]. Rearing substrates ‘Apple’ [19] and ‘Bread waste’ [28] both showed the lowest total lipid content of 0.0% of dry matter. The highest total lipid content in a substrate was reported for ‘Poultry waste’, with 42.9% lipid dry matter [25]. Rearing substrates ‘Melon’, ‘Tomato’ [19] and ‘Fish waste *O. mykiss*’ [28] all showed the lowest total carbohydrate content of 0.0% of dry matter. The highest total carbohydrate content of a rearing substrate was reported for ‘Bread’, with 78.6% carbohydrate dry matter [24].

Only one article reported rearing substrate macronutrient composition on a fresh weight basis, finding the highest protein and lipid content in substrate ‘Vegetable—lettuce, string green beans, cabbage (ratio of 3.3:3.3:3.3)’ (2.0% and 0.2% of fresh weight, respectively) and the highest substrate carbohydrate content in substrate ‘Fruit—apple, pear, orange (ratio of 3.3:3.3:3.3)’ (8.9% of fresh weight) [37].

### 3.5. Amino Acid Composition of Rearing Substrates

Two articles reported the amino acid content of the rearing substrate provided to the BSF larvae (Appendix A) [26,33]. Results collected indicated that glutamate was the most abundant amino acid found, accounting for 25.9% of total protein dry matter in ‘Brown algae *A. nodosum*’ [26]. No other substrate composition included this amino acid in isolation.

Cysteine and tryptophan were the least abundant amino acids reported in the rearing substrate with ‘Restaurant waste—potato, rice, pasta, vegetable (ratio unspecified)’ showing the presence of both amino acids at 0.2% of total protein dry matter [33].

### 3.6. Macronutrient Composition of Black Soldier Fly Larvae Reared on Food Waste

As per the inclusion criteria, all articles included in this review provided details on the protein composition of BSF larvae reared on food waste substrates. Of these, 22 articles reported BSF larvae protein composition on a dry matter basis, with one article reporting on a fresh weight basis (Table 3).

There was an average of 31.2% total protein when reported on a dry matter basis. Rearing substrate ‘Exotic fruit, pineapple, kiwi, apple, melon (ratio of 2:2:2:2:2)’ for 26.7 days, ‘Exotic fruit, kiwi (ratio of 5:5)’ for 25.3 days [19] and ‘Fruit and vegetable (uncharacterised)’ for an unspecified number of days [39], equally resulted in the lowest total protein content of 12.9% of BSF larvae dry matter. The highest BSF larvae total protein content was reported for larvae reared for two days on ‘Fish waste *S. aurita*’ with 78.8% total protein BSF larvae dry matter [40]. This was followed by ‘Fish waste *S. aurita*’ reared for one and four days and resulting in 77.4% and 75.4% total protein of BSF larvae dry matter, respectively [40].

Twelve articles reported additional details of the BSF larvae total lipid content on a dry matter basis [19,20,22,24,26,27,29,30,31,36,37,39,41], and two articles reported details of the BSF larvae total carbohydrate content, on a dry matter basis [39,41]. One article reported additional details of the BSF larvae total lipid content of a fresh weight basis [37].

Rearing substrates ‘Fruit and vegetables (uncharacterised)’ reared for an unspecified number of days [39], ‘Pomace, all-year mix (ratio of 5:5)’ reared for 26 days and ‘Pineapple’ reared for 33.7 days [19] resulted in the lowest reported total lipid content of 2.2%, 4.9% and 5.0% total lipid of BSF larvae dry matter, respectively. The highest BSF larvae total lipid content was reported for larvae reared for 14 days on ‘Bread’, resulting in 57.8% total lipid BSF larvae dry matter [24]. This was followed by ‘Spent barley, brewer’s yeast’ reared for an unspecified number of days [22] and ‘Fish *O. mykiss*, wheat (ratio of 5:1)’ reared for 14 days [24] (49.0% and 46.7% total lipid of BSF larvae dry matter, respectively).

Of the two articles reporting on BSF larvae total carbohydrate content (a combined total of three rearing substrates), ‘Fish waste (uncharacterised)’ larvae reared for 12 days produced the highest reported result of 12.7% total carbohydrate BSF larvae dry matter [40]. This [40] and ‘Fruit and vegetable (uncharacterised)’ reared for an unspecified number of days [39] (12.3% and 8.4% total carbohydrate of BSF larvae dry matter, respectively).

The one article reporting BSF larvae nutritional composition on a fresh weight basis showed the highest protein in BSF larvae reared on ‘Fruit and vegetable mix (ratio of 1:1)’ for 36.7 days (17.6% of fresh weight) and highest lipid content in BSF larvae reared on ‘Fruit—apple, pear, orange (ratio of 3.3:3.3:3.3)’ for 52 days (21.0% of fresh weight) [37].

### 3.7. Essential Amino Acid Composition of Black Soldier Fly Larvae Reared on Food Waste

Seven articles included in this review provided details on the essential amino acid composition of BSF larvae reared on different substrates. Of these, six articles reported BSF larvae essential amino acid composition on a dry matter basis [26,32,33,35,36,38], with one article reporting on a wet weight basis (Table 4) [41].

Histidine—Histidine comprised 2.5% of BSF larvae total protein content when reported on a dry matter basis. Rearing BSF larvae on ‘Kitchen waste—potato peelings, carrot, rice, bread debris (ratio unspecified)’ for an unspecified number of days, resulted in the lowest histidine content of 0.3% of the total larval protein content [32]. The highest BSF larvae histidine content was reported for BSF larvae reared for 15 days on ‘Carbohydrate—wheat middlings’ with 3.3% of BSF larvae total protein content [35].Isoleucine—Isoleucine comprised 3.8% of BSF larvae total protein content when reported on a dry matter basis. Rearing substrate ‘Brewery by-product spent grain’ reared for an unspecified number of days, resulted in the lowest reported isoleucine content of 0.2% of BSF larvae total protein content [32]. The highest BSF larvae isoleucine content was reported for BSF larvae reared for 43–47 days on ‘Fruit and vegetable mix—lettuce, apple, potato (5:3:2)’ with 4.3% of BSF larvae total protein content [38].Leucine—Leucine was the most abundant essential amino acid and comprised 6.3% of BSF larvae total protein content when reported on a dry matter basis. Rearing substrate ‘Kitchen waste—potato peelings, carrot, rice, bread debris (ratio unspecified)’ reared for an unspecified number of days, resulted in the lowest reported leucine content of 0.3% of BSF larvae total protein content [32]. The highest BSF larvae leucine content was reported for BSF larvae reared for eight days on ‘Wheat, brown algae *A. nodosum* (4:6)’ with 6.9% of BSF larvae total protein content [26].Lysine—Lysine comprised 5.6% of BSF larvae total protein content when reported on a dry matter basis. Rearing substrates ‘Kitchen waste—potato peelings, carrot, rice, bread debris (ratio unspecified)’ and ‘Brewery by-product spent grain’ reared for an unspecified number of days, both resulting in the lowest reported lysine content of 0.5% of BSF larvae total protein content [32]. The highest BSF larvae lysine content was reported for BSF larvae reared for 19 days on ‘Food waste (uncharacterised)’ with 8.3% of BSF larvae total protein content [38].Methionine—Methionine comprised 1.5% of BSF larvae total protein content when reported on a dry matter basis. Rearing substrates ‘Brewery by-product spent grain’ reared for an unspecified number of days [32] and ‘Restaurant waste—potato, rice, pasta, vegetables (ratio unspecified)’ reared for 19 days [33] both resulting in the lowest reported methionine content of 0.7% of BSF larvae total protein content. The highest BSF larvae methionine content was reported for BSF larvae reared for 45 days on ‘Fruit and vegetable mix—zucchini, apple, potato, green beans, carrot, pepper, orange, celery, kiwi, plum, eggplant (unspecified ratio)’ [36] and ‘Food waste (uncharacterised)’ reared for 19 days, both resulting in 1.8% methionine of BSF larvae total protein content [38].Phenylalanine—Phenylalanine comprised of 3.5% of BSF larvae total protein content when reported on a dry matter basis. Rearing substrate ‘Brewery by-product spent grain’ reared for an unspecified number of days, resulted in the lowest reported phenylalanine content of 0.2% of BSF larvae total protein content [32]. The highest BSF larvae phenylalanine content was reported for BSF larvae reared for eight days on ‘Wheat, brown algae *A. nodosum* (8:2)’ with 4.3% of BSF larvae total protein content [26].Threonine—Threonine comprised 3.8% of BSF larvae total protein content when reported on a dry matter basis. Rearing substrate ‘Restaurant waste—potato, rice, pasta, vegetables (ratio unspecified)’ reared for 19 days, resulted in the lowest reported threonine content of 1.6% of BSF larvae total protein content [33]. The highest BSF larvae threonine content was reported for BSF larvae reared for eight days on ‘Wheat, brown algae *A. nodosum* (8:2)’ and ‘Wheat, brown algae *A. nodosum* (4:6)’ both resulting in 4.1% threonine of BSF larvae total protein content [26].Tryptophan—Tryptophan was the least abundant essential amino acid and comprised 1.1% of BSF larvae total protein content when reported on a dry matter basis. Rearing substrate ‘Fruit and vegetable mix—zucchini, apple, potato, green beans, carrot, pepper, orange, celery, kiwi, plum, eggplant (unspecified ratio)’ reared for 45 days, resulted in the lowest reported tryptophan content of 0.4% of BSF larvae total protein content [36]. The highest BSF larvae tryptophan content was reported for BSF larvae reared for 19 days on ‘Food waste (uncharacterised)’ and BSF larvae reared for 42–47 days on ‘Fruit and vegetable mix—lettuce, apple, potato (5:3:2)’ both resulting in 1.4% tryptophan of BSF larvae total protein content [38].Valine—Valine comprised 5.5% of BSF larvae total protein content when reported on a dry matter basis. Rearing substrate ‘Kitchen waste—potato peelings, carrot, rice, bread debris (ratio unspecified)’ reared for an unspecified number of days, resulted in the lowest valine content of 0.1% of BSF larvae total protein content [32]. The highest BSF larvae valine content was reported for BSF larvae reared for eight days on ‘Wheat, brown algae *Ascophyllum nodosum* (9:1, 8:2 and 7:3)’ all resulting in 6.0% valine of BSF larvae total protein content [26].

The one article reporting BSF larvae essential amino acid composition on a wet weight basis showed the least and most abundant essential amino acid to be methionine (0.9% of BSF larvae total protein content) and valine (2.4% of BSF larvae total protein content), respectively, when reared on ‘Food waste (uncharacterised)’ for an unspecified number of days [41].

### 3.8. Non-Essential Amino Acid Composition of Black Soldier Fly Larvae Reared on Food Waste

Seven articles included in this review provided details on the non-essential amino acid composition of BSF larvae reared on different substrates. Of these, six articles reported BSF larvae non-essential amino acid composition on a dry matter basis [26,32,33,35,36,38], with one article reporting on a wet weight basis (Table 5) [41].

Alanine—Alanine comprised 6.2% of BSF larvae total protein content when reported on a dry matter basis. Rearing substrate ‘Restaurant waste—potato, rice, pasta, vegetable (ratio unspecified)’ reared for 19 days, resulted in the lowest reported alanine content of 2.8% of BSF larvae total protein content [33]. The highest BSF larvae alanine content was reported for BSF larvae reared for 15 days on ‘Carbohydrate—wheat middlings with 7.8% of BSF larvae total protein content [35].Arginine—Arginine comprised 4.3% of BSF larvae total protein content when reported on a dry matter basis. Rearing substrate ‘Brewery by-product spent grain’ reared for an unspecified number of days, resulted in the lowest reported arginine content of 0.3% of BSF larvae total protein content [32]. The highest BSF larvae arginine content was reported for BSF larvae reared for eight days on ‘Brown algae *A. nodosum*’ with 6.5% of BSF larvae total protein content [26].Aspartate—Aspartate comprised 8.1% of BSF larvae total protein content when reported on a dry matter basis. Rearing substrate ‘Restaurant waste—potato, rice, pasta, vegetable (ratio unspecified)’ reared for 19 days, resulted in the lowest reported aspartate content of 3.7% of BSF larvae total protein content [33]. The highest BSF larvae aspartate content was reported for BSF larvae reared for eight days on ‘Wheat’ with 9.4% of BSF larvae total protein content [26].Cysteine—Cysteine comprised 0.5% of BSF larvae total protein content when reported on a dry matter basis. Rearing substrate ‘Fruit and vegetable mix—zucchini, apple, potato, green beans, carrot, pepper, orange, celery, kiwi, plum, eggplant (unspecified ratio)’ reared for 45 days, resulted in the lowest reported cysteine content of 0.05% of BSF larvae total protein content [36]. The highest BSF larvae cysteine content was reported for BSF larvae reared for 15 days on ‘Carbohydrate—wheat middlings with 0.9% of BSF larvae total protein content [35].Glutamate—Of the one article (two rearing substrates) reporting BSF larvae glutamate composition, there was an average of 9.7% glutamate of BSF larvae total protein content when reported on a dry matter basis. Rearing substrate ‘Fruit and vegetable mix—lettuce, apple, potato (5:3:2)’ reared for 42–47 days, resulted in the lowest reported glutamate content of 9.5% of BSF larvae total protein content [38]. The highest BSF larvae glutamate content was reported for BSF larvae reared for 19 days on ‘Food waste (uncharacterised)’ with 9.8% of BSF larvae total protein content [38].Glutamic acid—Glutamic acid was the most abundant non-essential amino acid and comprised 9.8% of BSF larvae total protein content when reported on a dry matter basis. Rearing substrate ‘Brewery by-product spent grain’ reared for an unspecified number of days, resulted in the lowest reported glutamic acid content of 0.3% of BSF larvae total protein content [32]. The highest BSF larvae glutamic acid content was reported for BSF larvae reared for eight days on ‘Wheat, brown algae *A. nodosum* (4:6)’ with 12.8% of BSF larvae total protein content [26].Glutamine—One article (two rearing substrates) reporting BSF larvae glutamine composition, there was an average of 0.4% glutamine of BSF larvae total protein content when reported on a dry matter basis and was the least abundant non-essential amino acid. Rearing substrate ‘Brewery by-product spent grain’ for an unspecified number of days, resulted in the lowest reported glutamine content of 0.0% of BSF larvae total protein content. The highest BSF larvae glutamine content was reported for BSF larvae reared for an unspecified number of days on ‘Kitchen waste—potato peelings, carrot, rice, bread debris (ratio unspecified)’ with 0.8% of BSF larvae total protein content [32].Glycine—Glycine comprised 4.8% of BSF larvae total protein content when reported on a dry matter basis. Rearing substrate ‘Fruit and vegetable mix—zucchini, apple, potato, green beans, carrot, pepper, orange, celery, kiwi, plum, eggplant (unspecified ratio)’ reared for 45 days, resulted in the lowest reported glycine content of 2.3% of BSF larvae total protein content [36]. The highest BSF larvae glycine content was reported for BSF larvae reared for 15 days on ‘Carbohydrate—wheat middlings with 5.6% of BSF larvae total protein content [35].Proline—Proline comprised 4.6% of BSF larvae total protein content when reported on a dry matter basis. Rearing substrate ‘Brewery by-product spent grain’ reared for an unspecified number of days, resulted in the lowest reported proline content of 0.2% of BSF larvae total protein content [32]. The highest BSF larvae proline content was reported for BSF larvae reared for 15 days on ‘Carbohydrate—wheat middlings with 6.2% of BSF larvae total protein content [35].Serine—Serine comprised 4.1% of BSF larvae total protein content when reported on a dry matter basis. Rearing substrate ‘Restaurant waste—potato, rice, pasta, vegetable (ratio unspecified)’ reared for 19 days, resulted in the lowest reported serine content of 1.6% of BSF larvae total protein content [33]. The highest BSF larvae serine content was reported for BSF larvae reared for eight days on ‘Wheat, brown algae *A. nodosum* (4:6)’ with 4.9% of BSF larvae total protein content [26].Tyrosine—Tyrosine comprised 4.1% of BSF larvae total protein content when reported on a dry matter basis. Rearing substrate ‘Brewery by-product spent grain’ reared for an unspecified number of days, resulted in the lowest reported tyrosine content of 0.3% of BSF larvae total protein content [32]. The highest BSF larvae tyrosine content was reported for BSF larvae reared for 19 days on ‘Food waste (uncharacterised)’ with 6.0% of BSF larvae total protein content [38].

The one article reporting BSF larvae non-essential amino acid composition on a wet weight basis showed the least and most abundant non-essential amino acid to be cysteine (1.1% of BSF larvae total protein content) and alanine (2.7% of BSF larvae total protein content), respectively, when reared on ‘Food waste (uncharacterised)’ for an unspecified number of days [41].

## 4. Risk of Bias Assessment

Risk of Bias (ROB) assessment was performed using the Systematic Review Centre for Laboratory Animal Experimentation (SYRCLE) risk of bias tool [18], with the exclusion of items 5 and 7 (Appendix A). These items were removed from assessment following the instruction of the (SYRCLE) risk of bias tool to adapt the list to the specific needs of the review [18], with ‘blinding of the caregiver’ and ‘blinding of the assessors’ deemed by the authors as unlikely to influence the potential bias of the articles reviewed. This ROB assessment performed in this review highlights a potential widespread nature of poor method reporting and lack of intent to reduce the risk of bias in BSF larvae rearing investigations. All articles included in this review were missing information to various degrees, with all showing potential for selection bias due to the absence of methodological information reporting on allocation concealment or sequencing generation and as such, being at risk of a significant level of bias.

## 5. Discussion

BSF larvae offer a feasible and cost-effective solution to two growing global challenges: food waste management and the rising global demand for sustainable protein. For this decomposer insect to be utilised in the treatment of food waste, and then to be effectively implemented and accepted into the food supply, it is essential to further our knowledge regarding the influence of various food waste streams on the nutritional composition of the BSF larvae.

Dietary protein is an essential macronutrient in BSF larval development and is necessary for supporting adequate protein and lipid accumulation in the fat cells of the BSF [46]. The results of this review indicate the total protein content of the BSF larvae can be substantially influenced by the substrate protein content of which it is reared on, with BSF larvae reared for up to 28.7 days on various combinations of low protein ‘Fruit and vegetable’ substrates producing the least abundant source of BSF larval protein (12.9% total protein) [19,39]. In comparison, BSF larvae reared for two days on a high protein substrate ‘Fish waste *S. aurita*’ produced the most abundant source of BSF larval protein (78.8% total protein) [40].

Interestingly, BSF larvae reared on ‘Fish waste *S. aurita*’ for longer than two days displayed a steady decline in larval total protein content, suggesting that two days of rearing on this high protein food waste substrate would be the optimal condition to generate high protein BSF larvae [40]. Variation in the protein composition of the BSF larvae throughout their lifespan has been supported by others using non-food waste rearing substrates (Chicken feed), in which total protein has been reported to range between 34% and 42% during larval stages and between 31% and 46% [47,48,49,50]. The articles collated in this review began experimental feeding procedures at various stages in the BSF larvae life cycle, from placing eggs directly onto the rearing diets [22,36] to waiting until the BSF larvae were aged up to 14 days before introducing them to food waste substrates [25]. Furthermore, some studies reported different harvest stages, including the sight of first prepupae [30], when 40% of BSF larvae had reached prepupae [36,37] and when 100% of BSF larvae had reached prepupae [38] (Appendix A). As such, the composition of total protein in the BSF larvae reported across different studies may be compounded by factors such as age and harvest stage, rather than a representation of the type of food waste provisions.

The articles included in this review also reported widespread difference in the quantity of feed rations provided to the BSF larvae, with a range from 12.2 mg per larvae [23] to 1530 mg per larvae [24]. Insufficient feed rations have been shown to impair biomass production and influence BSF larvae protein content [48]. Whilst the optimal feeding ration of both chicken feed and fecal sludge has been determined for optimising BSF larvae biomass and nutritional content, the literature on the ideal food waste provision is limited [51]. This is likely due to the variations and inconsistencies in food waste products from a macronutrient perspective. It is plausible that the BSF larvae in the articles included in this review, may have been provided with less than adequate feed rations and as a result their protein composition was influenced by lack of sustenance as opposed to the macronutrient content of the rearing substrate provided.

In addition to rearing conditions influencing the composition of total protein in the BSF larvae, different data analyses methods used by different studies may have impacted on the ability to accurately compare the efficacy of differing food waste substrates. The total protein content of BSF larvae is most commonly determined from the total elemental nitrogen content using the common Kjeldahl method and the standard protein conversion factor 6.25 [52]. However, due to the additional non-protein nitrogen found in the insects’ chitin, it is possible to over-estimate total protein content and as such a factor of 4.67 has been proposed as a more accurate representation of total protein content in insects [52]. When presenting the proximate composition of BSF larvae, Ewald et al. [24] included results of both conversion factors highlighting the differences between both factors. BSF larvae reared on ‘Bread’ for 14 days, with protein determination calculated with a conversion factor of 6.25, indicated the substrate to result in a total protein content of 39.2% of BSF larvae dry matter [24]. Yet, when the same data were reanalysed using a conversion factor of 4.67, a total protein content of 29.8% of BSF larvae dry matter was indicated; a substantial difference of 9.4% in the total protein content reported in BSF larvae [24]. Spranghers et al. [33] also included chitin corrected values when observing the influence of ‘Restaurant waste—potato, rice, pasta, vegetable (ratio unspecified)’ on the nutritional composition of the BSF, finding a similar decrease of 9.0% in BSF larvae total protein content when compared to data not taking chitin into consideration [33]. It is possible to include chitin corrected value, as this is obtained by analysing the nitrogen content of the chitin fraction and subtracting this from total nitrogen content, yet the reporting of the conversion factors was absent in many articles included in this review [20,21,23,36,37,39], as was the reporting of chitin correction adjustments [19,20,21,22,24,25,26,27,28,30,31,32,34,35,37,38,39,40,41], which may hinder the ability to draw comparisons between the total protein content of BSF larvae reared on various food waste substrates included in this review.

The BSF larvae’s amino acid profile has been shown to be suitable for use as pet food [47] and animal feed [41]. To date, only a few studies have examined the relationship between the substrate amino acid composition and that of the BSF larvae [25,32]. When observing the impact of food waste substrates on the amino acid profiles of BSF larvae, of the two articles included in this review, glutamic acid was reported to be the most abundant non-essential amino acid (when reared on ‘Wheat, brown algae *A. nodosum* (4:6)’ for eight days) [26] and leucine the most abundant essential amino acid (when reared on ‘Wheat, brown algae *A. nodosum* (4:6)’ for eight days) [26]. This was consistent with the glutamic acid and leucine content also being reported as most prevalent in the food waste rearing substrates provided to BSF larvae [26,33]. Aside from both glutamic acid and leucine, there was a great variation in the amino acid content of substrates used in different studies (Appendix A.). Despite this, the BSF larvae only exhibited minimal differences in amino acid content (±20%) within each study [33,38]. This would indicate that the amino acid content of the BSF larvae has a limited opportunity for manipulation when reared on food waste products regardless of the amino acid content of the rearing substrate provided. With only two articles providing amino acid content of both BSF larvae and the rearing substrate, there is limited information available to draw a solid conclusion on the role of rearing substrates in the influence of the amino acid content on BSF larvae. This makes further studies essential to developing a clearer understand of this relationship.

The choice of processing methods has been shown to influence the amino acid content of the BSF larvae, including the culling and drying method used in preparing the BSF larvae [53]. Articles included in this review reported various drying techniques including freezing [23,34] and freeze-drying in liquid nitrogen [29]. Huang et al. [53] reported that conventional drying (60.0 °C) of BSF larvae produced a higher digestible indispensable amino acid score when compared with microwave drying of BSF larvae. Others have reported that culling BSF larvae by a method of freezing resulted in a reduction in amino acids cysteine and lysine content [54]. Both Liland et al. [26] and Spranghers et al. [33] reporting freezing as their method of culling BSF larvae, there is an increased likelihood of the amino acid content values of the BSF larvae being inaccurately reported.

In addition to rearing substrate, various abiotic factors can affect the development of BSF larvae and may explain the variability of the nutritional content of BSF larvae in the studies included in this review. Abiotic factors that may influence BSF larvae development include larval density [55], larval handling [42], substrate moisture content or pH level [56], however, these factors were not extracted from articles and taken into consideration in this current review.

## 6. Conclusions and Future Directions

The results of this review on the influence of food waste products on BSF larvae protein content infer that the total protein content of food waste products used as a rearing substrate is likely to result in producing BSF larvae with a similar total protein and amino acid content. However, due to the variation in methodologies applied within each study and absence of BSF larvae nutritional composition at the commencement of the studies, there is a reduced confidence in the extent to which these various food waste substrates may have influenced the total protein content of BSF larvae. The standardisation of methodologies in BSF larvae resource conversion studies have been proposed by others and adherence to a standard methodology may increase confidence in future studies [44].

The transformation and nutrient recovery prospects of using BSF larvae as a food waste management system and as a sustainable protein source are promising. However, further research is required regarding the influence of various food waste streams on the protein composition outcomes of the BSF larvae, as well as a greater understanding of the potential influence of anti-nutritive elements on the nutritional profiles of BSF larvae. Further research exploring these factors will improve the successful introduction of BSF larvae as a novel feed and/or food.

## Figures and Tables

**Figure 1 insects-12-00608-f001:**
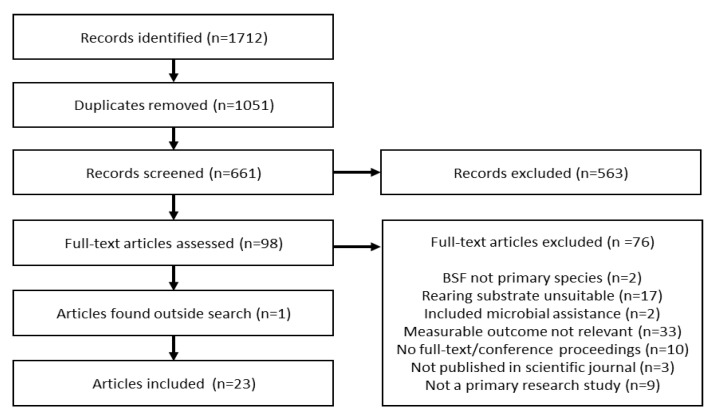
Flow diagram summarising the screening process.

**Table 1 insects-12-00608-t001:** Rearing substrates (RS) and rearing conditions of the BSF larvae.

Author	Rearing Substrate (RS) (Mixture Ratio)	Rearing Duration (Days)	Feed Ration (mg Per Larvae)	Frequency of Feed	Larval Age at Day 1 (Days)	Temp (°C)	Relative Humidity (%)	Light: Darkness (Hours)
Barbi [19]	RS 1–21: Exotic fruit, pineapple, kiwi, apple, melon (various combinations)	12–47	250–375	-	2–3 instar	27.0 ± 0.5	60.0–70.0	16:8
	RS 22–34: All-year mix, peach, tomato (various combinations)							
	RS 35–49: Legume, corn, pomace, all-year mix (various combinations)							
Barragán-Fonseca [20]	RS 1: Low protein—dried distillers’ grains with soluble, cabbage leaves, old bread, cellulose, sunflower oil (unspecified ratio)	21–22	1000	Once at beginning	1	27.0 ± 1.0	70.0 ± 5.0	12:12
	RS 2: High protein—dried distillers’ grains with soluble, cabbage leaves, old bread, cellulose, sunflower oil (unspecified ratio)							
Bava [21]	RS 1: Okara	16–22	-	Ab libitum	2	25.0 ± 0.5	60.0 ± 0.5	12:12
	RS 2: Maize distillers							
	RS 3: Brewers’ grains							
Cappellozza [36]	RS 1: Fruit and vegetable mix—zucchini, apple, potato, green beans, carrot, pepper, orange, celery, kiwi, plum, eggplant (unspecified ratio)	45	120	Daily after 10 days	Egg	27.0 ± 1.0	50.0 ± 0.5	16:8
Chia [22]	RS 1: Spent barley	14–21	≈1000	Every 3 days	Egg	28.0 ± 1.0	70.0 ± 2.0	-
	RS 2: Spent barley, brewer’s yeast							
	RS 3: Spent barley, brewer’s yeast, molasses							
	RS 4: Spent malted barley							
	RS 5: Spent malted barley, brewer’s yeast							
	RS 6: Spent malted barley, brewer’s yeast, molasses							
	RS 7: Spent malted corn							
	RS 8: Spent malted corn, brewer’s yeast							
	RS 9: Spent malted corn, brewer’s yeast, molasses							
	RS 10: Spent sorghum, barley							
	RS 11: Spent sorghum, barley, brewer’s yeast							
	RS 12: Spent sorghum, barley, brewer’s yeast, molasses							
Danieli [23]	RS 1: Control—ground corn, wheat bran, dehydrated alfalfa (5:2:3)	21	12.2	Every 2–3 days	6	28.5 ± 0.3	75.6 ± 4.2	-
	RS 2: High non-fibre carbohydrate—ground barley, wheat bran, dehydrated alfalfa (6.8:2:1.2)							
	RS 3: High fibre carbohydrate—ground barley, wheat middlings, dehydrated alfalfa, wheat straw (1.6:5:1:2.4)							
	RS 4: High protein—ground barley, wheat middlings, dehydrated alfalfa (1.5:5.5:3)							
Ewald [24]	RS 1: Bread	14	250	-	5	28.0	-	-
	RS 2: Fish *O. mykiss*, wheat (5:1)		230					
	RS 3: Food waste (uncharacterised)		170					
	RS 4: Fresh mussels *M. edulis*		1530					
	RS 5–9: Bread, fresh mussels *M. edulis* (9:1,8:2,7:3,6:4,5:5)		280–300					
Gold [25]	RS 1: Mill by-products	9	15–40	Every 3 days	12–14	28.0	70.0	-
	RS 2: Canteen waste—mix of vegetables with/without dressing, sausage, offal (ratio unspecified)							
	RS 3: Poultry waste							
	RS 4: Vegetable canteen waste—mix of vegetables with/without dressing (ratio unspecified)							
	RS 5: Mixed food waste—(1:1:1 of RS1:RS2:RS3)							
Liland [26]	RS 1: Wheat	8	Various as per consumption rates	Daily	8	30.0	65.0	0:24
	RS 2–10: Wheat, brown algae *A. nodosum* (9:1, 8:2, 7:3, 6:4, 5:5, 4:6, 3:7, 2:8, 1:9)							
	RS 11: Brown algae *A. nodosum*							
Liu [27]	RS 1: Brewery by-products (uncharacterised)	15	200	Every 5 days	7	24.5 ± 1.5	40.0 ± 10.0	12:12
Lopes [28]	RS 1: Bread	11–12	250	On days 0,4,7	7	28.0 ± 1.5	45.0 ± 6.3	-
	RS 2–4: Bread, fish *O. mykiss* (9.5:0.5,9:1,8.5:1.5)							
Meneguz [29]	RS 1: Fruit and vegetable mix—celery, oranges, peppers (4.3:2.9:2.8)	23–28	1000	Ad libitum	6	27.0 ± 5.0	70.0 ± 5.0	24:0
	RS 2: Fruit—apples, oranges, apple leftovers, strawberries, mandarins, pears, kiwis, bananas, lemons (4.8:1.5:1.4:0.7:0.5:0.4:0.3:0.2:0.2)							
	RS 3: Winery by-products—grape seeds, pulp, skins, stems, leaves (ratio unspecified)							
	RS 4: Brewery by-products—barley brewers’ grains							
Oonincx [30]	RS 1: High protein high fat—spent grains, beer yeast, cookie remains (6:2:2)	21–37	40	Ad libitum	<1	28.0	70.0	12:12
	RS 2: High protein low fat—beer yeast, potato stem peelings, beet molasses (5:3:2)							
	RS 3: Low protein high fat—cookie remains, bread (5:5)							
	RS 4: Low protein low fat—potato steam peelings, beet molasses, bread (3:2:5)							
Salomone [31]	RS 1: Food waste—vegetable, meat/fish, bread/pasta/rice, other (6.5:0.5:2.5:0.5)	12	-	-	-	30–35	>65.0	-
Shumo [32]	RS 1: Kitchen waste—potato peelings, carrot, rice, bread debris (ratio unspecified)	21–28	-	Weekly	Neonate	28.0 ± 2.0	65.0 ± 5.0	-
	RS 2: Brewery by-product spent grain							
Spranghers [33]	RS 1: Restaurant waste—potato, rice, pasta, vegetables (ratio unspecified)	15–19	600	Every 3 days	6–8	27.0 ± 1.0	65.0 ± 5.0	-
Tinder [34]	RS 1: Sorghum	28–38	33.3	Daily	4	28.0 ± 2.0	70.0	14:10
	RS 2–4: Sorghum, cowpea (7.5:2.5, 5:5, 2.5:7.5)							
	RS 5: Cowpea							
Tschirner [35]	RS 1: Carbohydrate—wheat middlings	15	-	Once at beginning	8	28.0	73.8–75.7	-
	RS 2: Protein—dried distillers’ grains with soluble							
	RS 3: Fibre—sugar beet							
Cappellozza [36]	RS 1: Fruit and vegetable mix—zucchini, apple, potato, green beans, carrot, pepper, orange, celery, kiwi, plum, eggplant (unspecified ratio)	45	120	Daily after 10 days	Egg	27.0 ± 1.0	50.0 ± 0.5	16:8
Jucker [37]	RS 1: Fruit—apple, pear, orange (3.3:3.3:3.3)	37–52	-	Ad libitum	9	25.0 ± 0.5	60.0 ± 0.5	12:12
	RS 2: Vegetable—lettuce, string green beans, cabbage (3.3:3.3:3.3)							
	RS 3: Fruit and vegetable mix—(1:1 of RS1:RS2)							
Lalander [38]	RS 1: Food waste (uncharacterised)	19–47	40	Every 2–3 days	10	28.0	-	-
	RS 2: Fruit and vegetable mix—lettuce, apple, potato (5:3:2)							
Nguyen [39]	RS 1: Kitchen waste (animal and plant matter)	19–40	40	Ad libitum	4	28.0	60.0 ± 10.0	-
	RS 2: Fruit and vegetable (uncharacterised)							
	RS 3: Fish (uncharacterised)							
Barroso [40]	RS 1: Fish *S. aurita*	1–12	-	Ab libitum	3	26.0 ± 1.0	65.0 ± 5.0	-
Surendra [41]	RS 1: Food waste (uncharacterised)	-	-	-	-	-	-	-

≈, approximately; g, grams; mg, milligrams; C, Celsius. Nguyen et al. [42] article data used to support Nguyen et al. [39] reporting of rearing conditions. Second or third stage instar implies ≈ 15 days of age [43]. Neonate implies <5 days of age [44].

**Table 2 insects-12-00608-t002:** Macronutrient Composition of Rearing Substrates (RS).

Author	Rearing Substrate (RS) (Mixture Ratio)	Rearing Substrate (RS)					
		Protein (%)		Lipid (%)		Carbohydrate (%)	
		DM	FW	DM	FW	DM	FW
Barbi [19]	RS 1: Exotic fruit	0.7	-	0.1	-	15.4	-
	RS 2: Pineapple	0.5	-	0.2	-	11.4	-
	RS 3: Kiwi	0.9	-	0.1	-	7.4	-
	RS 4: Apple	0.4	-	0.0	-	6.9	-
	RS 5: Melon	0.5	-	0.1	-	0.0	-
	RS 6: Tomato	3.0	-	0.4	-	0.0	-
	RS 7: Peach	0.9	-	0.8	-	4.3	-
	RS 8: Pomace	2.0	-	3.7	-	8.2	-
	RS 9: Legume	17.0	-	0.7	-	6.3	-
	RS 10: Corn	11.5	-	1.4	-	0.3	-
Barragán–Fonseca [20]	RS 1: Low protein—dried distillers’ grains with soluble, cabbage leaves, old bread, cellulose and sunflower oil (ratio unspecified)	10.0	-	-	-	30.0	-
	RS 2: High protein—dried distillers’ grains with soluble, cabbage leaves, old bread, cellulose and sunflower oil (ratio unspecified)	17.0	-	-	-	30.0	-
Bava ^~^ [21]	RS 1: Maize distillers	39.2	-	-	-	7.5	-
	RS 2: Okara	29.5	-	-	-	17.3	-
	RS 3: Brewer’s grains	15.8	-	-	-	11.2	-
Chia [22]	RS 1: Spent barley	30.3	-	6.4	-	-	-
	RS 2: Spent barley, brewer’s yeast (3.6:6.4)	32.0	-	5.4	-	-	-
	RS 3: Spent barley, brewer’s yeast, molasses (3.6:3.2:3:2)	22.1	-	4.0	-	-	-
	RS 4: Spent malted barley	28.9	-	6.8	-	-	-
	RS 5: Spent malted barley, brewer’s yeast (3.6:6.4)	30.2	-	7.0	-	-	-
	RS 6: Spent malted barley, brewer’s yeast, molasses (3.6:3.2:3:2)	22.3	-	3.2	-	-	-
	RS 7: Spent malted corn	27.4	-	6.5	-	-	-
	RS 8: Spent malted corn, brewer’s yeast (3.6:6.4)	27.7	-	6.0	-	-	-
	RS 9: Spent malted corn, brewer’s yeast, molasses (3.6:3.2:3:2)	19.1	-	3.4	-	-	-
	RS 10: Spent sorghum, barley	29.4	-	11.8	-	-	-
	RS 11: Spent sorghum, barley, brewer’s yeast (3.6:6.4)	31.4	-	9.5	-	-	-
	RS 12: Spent sorghum, barley, brewer’s yeast, molasses (3.6:3.2:3:2)	21.7	-	5.2	-	-	-
Danieli ^^^ [23]	RS 1: Control—ground corn, wheat bran, dehydrated alfalfa (5:2:3)	10.6	-	-	-	59.7	-
	RS 2: High non-fibre carbohydrate—ground barley, wheat bran, dehydrated alfalfa (6.8:2:1.2)	11.1	-	-	-	68.9	-
	RS 3: High fibre carbohydrate—ground barley, wheat middlings, dehydrated alfalfa, wheat straw (1.6:5:1:2.4)	11.2	-	-	-	51.1	-
	RS 4: High protein—ground barley, wheat middlings, dehydrated alfalfa (1.5:5.5:3)	13.8	-	-	-	56.1	-
Ewald [24]	RS 1: Bread	13.5	-	5.3	-	78.6	-
	RS 2: Fish *O. mykiss*, wheat (5:1)	41.8	-	22.5	-	27.8	-
	RS 3: Food waste (uncharacterised)	20.5	-	20.7	-	48.4	-
	RS 4: Fresh mussels *M. edulis*	17.4	-	1.4	-	7.2	-
	RS 5: Bread, fresh mussels *M. edulis* (9:1)	14.5	-	3.0	-	56.9	-
	RS 6: Bread, fresh mussels *M. edulis* (8:2)	15.3	-	2.3	-	42.9	-
	RS 7: Bread, fresh mussels *M. edulis* (7:3)	15.8	-	1.9	-	33.7	-
	RS 8: Bread, fresh mussels *M. edulis* (6:4)	16.2	-	1.6	-	27.1	-
	RS 9: Bread, fresh mussels *M. edulis* (5:5)	16.4	-	1.3	-	22.2	-
Gold [25]	RS 1: Milled by-products	14.5	-	3.0	-	23.3	-
	RS 2: Canteen waste—mix of vegetables with/without dressing, sausage, offal (ratio unspecified)	32.2	-	34.9	-	7.5	-
	RS 3: Poultry waste	37.3	-	42.9	-	0.3	-
	RS 4: Vegetable canteen waste—mix of vegetables with/without dressing (ratio unspecified)	12.1	-	28.9	-	15.5	-
	RS 5: Mixed food waste—(1:1:1 of RS1:RS2:RS3)	19.6	-	22.3	-	15.4	-
Liland [26]	RS 1: Wheat	10.8	-	4.8	-	-	-
	RS 2: Wheat, brown algae *A. nodosum* (9:1)	9.8	-	4.8	-	-	-
	RS 3: Wheat, brown algae *A. nodosum* (8:2)	9.6	-	4.2	-	-	-
	RS 4: Wheat, brown algae *A. nodosum* (7:3)	8.6	-	3.3	-	-	-
	RS 5: Wheat, brown algae *A. nodosum* (6:4)	8.5	-	3.3	-	-	-
	RS 6: Wheat, brown algae *A. nodosum* (5:5)	7.4	-	3.3	-	-	-
	RS 7: Wheat, brown algae *A. nodosum* (4:6)	6.7	-	4.0	-	-	-
	RS 8: Wheat, brown algae *A. nodosum* (3:7)	6.5	-	2.8	-	-	-
	RS 9: Wheat, brown algae *A. nodosum* (2:8)	5.1	-	2.1	-	-	-
	RS 10: Wheat, brown algae *A. nodosum* (1:9)	5.3	-	2.4	-	-	-
	RS 11: Brown algae *A. nodosum*	4.5	-	2.0	-	-	-
Liu* [27]	RS 1: Brewery by-product (uncharacterised)	22.6	-	5.8	-	8.9	-
Lopes [28]	RS 1: Bread waste	8.2	-	0.0	-	46.1	-
	RS 2: Fish waste *O. mykiss*	60.3	-	32.5	-	0.0	-
Meneguz ^^^ [29]	RS 1: Fruit and vegetable mix—celery, oranges, peppers (4.3:2.9:2.8)	12.0	-	2.1	-	58.5	-
	RS 2: Fruit-apples, oranges, apple leftovers, strawberries, mandarins, pears, kiwis, bananas, lemons (4.8:1.5:1.4:0.7:0.5:0.4:0.3:0.2:0.2)	4.6	-	1.0	-	75.7	-
	RS 3: Winery by-products—grape seeds, pulp, skins, stems, leaves (ratio unspecified)	11.7	-	7.4	-	13.4	-
	RS 4: Brewery by-products—barley brewers’ grains	20.1	-	8.2	-	22.6	-
Oonincx [30]	RS 1: High protein high fat—spent grains, beer yeast, cookie remain (6:2:2)	21.9	-	8.9	-	-	-
	RS 2: High protein low fat—beer yeast, potato steam peelings, beet molasses (5:3:2)	22.9	-	0.4	-	-	-
	RS 3: Low protein high fat—cookie remains, bread (5:5)	12.9	-	14.0	-	-	-
	RS 4: Low protein low fat—potato steam peelings, beet molasses, bread (3:2:5)	14.4	-	1.5	-	-	-
Salomone [31]	RS 1: Food waste—vegetable, meat/fish, bread/pasta/rice, other (6.5:0.5:2.5:0.5)	-	-	-	-	-	-
Shumo [32]	RS 1: Kitchen waste—potato peelings, carrot, rice, bread debris (ratio unspecified)	20.0	-	-	-	-	-
	RS 2: Brewery by-product spent grain	12.2	-	-	-	-	-
Spranghers ^^^ [33]	RS 1: Restaurant waste—potato, rice, pasta, vegetables (ratio unspecified)	15.7	-	-	-	61.8	-
Tinder *^~^ [34]	RS 1: Sorghum	3.5	-	1.0	-	23.7	-
	RS 2: Cowpea	7.7	-	0.5	-	20.8	-
Tschirner [35]	RS 1: Carbohydrate—wheat middlings	22.0	-	-	-	-	-
	RS 2: Protein—dried distillers’ grains with soluble	31.2	-	-	-	-	-
	RS 3: Fibre—sugar beet	8.5	-	-	-	-	-
Cappellozza [36]	RS 1: Fruit and vegetable mix—zucchini, apple, potato, green beans, carrot, pepper, orange, celery, kiwi, plum, eggplant (unspecified ratio)	-	-	-	-	-	-
Jucker *^~^ [37]	RS 1: Fruit—apple, pear, orange (3.3:3.3:3.3)	-	0.4	-	0.1	-	8.9
	RS 2: Vegetable—lettuce, string green beans, cabbage (3.3:3.3:3.3)	-	2.0	-	0.2	-	2.4
	RS 3: Fruit and vegetable mix—(1:1 of RS1:RS2)	-	1.2	-	0.2	-	5.6
Lalander [38]	RS 1: Food waste (uncharacterised)	22.2	-	-	-	55.0 ^~^	-
	RS 2: Fruit and vegetable mix—lettuce, apple, potato (5:3:2)	13.2	-	-	-	72.6 ^~^	-
Nguyen * [39]	RS 1: Kitchen waste (animal and plant matter)	20.4	-	19.6	-	56.8	-
	RS 2: Fruit and vegetable (uncharacterised)	20.1	-	1.6	-	69.0	-
	RS 3: Fish (uncharacterised)	50.0	-	36.2	-	0.6	-
Barroso [40]	RS 1: Fish waste *S. aurita*	72.7	-	-	-	-	-
Surendra [41]	RS 1: Food waste (uncharacterised)	-	-	-	-	-	-

Results presented as a percentage. * Indicative of original article presenting data as g/100 g of rearing substrate. ^^^ indicative of original article presenting data as g/kg of rearing substrate. ^~^ indicative of original article nutrient data acquired from database or literature. Dashes used to indicate where data is unreported in the original article. Chia et al. [45] article data used to support Chia et al. [22] reporting of rearing substrate composition. DM dry matter, FW fresh weight.

**Table 3 insects-12-00608-t003:** Macronutrient Composition of Black Solider Fly Larvae Reared on Food Waste.

Author	Rearing Substrate (RS) (Mixture Ratio)	Black Soldier Fly Larvae					
		Protein (%)		Lipid (%)		Carbohydrate (%)	
		DM	FW	DM	FW	DM	FW
Barbi [19]	RS 1: Exotic fruit, melon (5:5)	14.0	-	6.5	-	-	-
	RS 2: Exotic fruit, pineapple, kiwi, apple, melon (1:1:6:1:1)	13.5	-	7.7	-	-	-
	RS 3: Pineapple	13.4	-	5.0	-	-	-
	RS 4: Melon	15.1	-	5.1	-	-	-
	RS 5: Apple	13.9	-	7.0	-	-	-
	RS 6: Exotic fruit	13.8	-	9.6	-	-	-
	RS 7: Exotic fruit, pineapple, kiwi, apple, melon (2:2:2:2:2)	12.9	-	8.8	-	-	-
	RS 8: Exotic fruit, kiwi (5:5)	12.9	-	8.4	-	-	-
	RS 9: Pineapple, melon (5:5)	13.9	-	6.0	-	-	-
	RS 10: Kiwi, melon (5:5)	13.5	-	6.0	-	-	-
	RS 11: Pineapple, apple (5:5)	13.9	-	7.7	-	-	-
	RS 12: Exotic fruit, pineapple, kiwi, apple, melon (1:1:1:6:1)	14.2	-	7.3	-	-	-
	RS 13: Apple, melon (5:5)	14.3	-	7.5	-	-	-
	RS 14: Exotic fruit, pineapple, kiwi, apple, melon (1:1:1:1:6)	13.9	-	5.8	-	-	-
	RS 15: Pineapple, kiwi (5:5)	14.5	-	7.6	-	-	-
	RS 16: Exotic fruit, pineapple (5:5)	14.1	-	8.8	-	-	-
	RS 17: Kiwi	13.1	-	7.9	-	-	-
	RS 18: Exotic fruit, apple (5:5)	13.9	-	9.5	-	-	-
	RS 19: Exotic fruit, pineapple, kiwi, apple, melon (6:1:1:1:1)	13.6	-	8.5	-	-	-
	RS 20: Exotic fruit, pineapple, kiwi, apple, melon (1:6:1:1:1)	14.2	-	7.5	-	-	-
	RS 21: Kiwi, apple (5:5)	13.1	-	7.0	-	-	-
	RS 22: Peach, tomato (6.7:3.3)	15.7	-	13.7	-	-	-
	RS 23: Peach	15.5	-	13.6	-	-	-
	RS 24: All-year mix, peach, tomato (6.7:1.6:1.7)	15.0	-	12.1	-	-	-
	RS 25: All-year mix, peach, tomato (3.4:3.3:3.3)	15.3	-	12.1	-	-	-
	RS 26: All-year mix, tomato (3.3:6.7)	15.6	-	10.6	-	-	-
	RS 27: All-year mix, peach (6.7:3.3)	14.6	-	10.7	-	-	-
	RS 28: All-year mix, peach, tomato (1.7:6.7:1.6)	16.4	-	14.2	-	-	-
	RS 29: All-year mix, tomato (6.7:3.3)	15.3	-	7.5	-	-	-
	RS 30: All-year mix	14.8	-	9.0	-	-	-
	RS 31: Peach, tomato (3.3:6.7)	14.8	-	12.2	-	-	-
	RS 32: All-year mix, peach, tomato (1.6:1.7:6.7)	15.9	-	12.6	-	-	-
	RS 33: Tomato	15.7	-	11.4	-	-	-
	RS 34: All-year mix, peach (3.3:6.7)	16.0	-	12.5	-	-	-
	RS 35: Legume, corn, pomace, all-year mix (1.25:6.25:1.25:1.25)	16.9	-	12.0	-	-	-
	RS 36: Corn, all-year mix (5:5)	16.9	-	11.6	-	-	-
	RS 37: Corn	18.4	-	10.2	-	-	-
	RS 38: Legume, corn, pomace, all-year mix (1.25:1.25:6.25:1.25)	16.5	-	8.7	-	-	-
	RS 39: Legume, corn, pomace, all-year mix (6.25:1.25:1.25:1.25)	16.2	-	11.0	-	-	-
	RS 40: Legume	17.3	-	7.0	-	-	-
	RS 41: Legume, pomace (5:5)	16.5	-	6.2	-	-	-
	RS 42: All-year mix	15.3	-	8.9	-	-	-
	RS 43: Legume, corn (5:5)	17.6	-	7.4	-	-	-
	RS 44: Pomace, all-year mix (5:5)	14.8	-	4.9	-	-	-
	RS 45: Legume, corn, pomace, all-year mix (1.25:1.25:1.25:6.25)	17.8	-	7.6	-	-	-
	RS 46: Pomace	14.8	-	5.6	-	-	-
	RS 47: Corn, pomace (5:5)	17.4	-	7.8	-	-	-
	RS 48: Legume, all-year mix (5:5)	16.9	-	7.7	-	-	-
	RS 49: Legume, corn, pomace, all-year mix (2.5:2.5:2.5:2.5)	17.7	-	8.7	-	-	-
Barragán- Fonseca ^≈^ [20]	RS 1: Low protein—dried distillers’ grains with soluble, cabbage leaves, old bread, cellulose, sunflower oil (ratio unspecified)	47.0	-	20.0	-	-	-
	RS 2: High protein—dried distillers’ grains with soluble, cabbage leaves, old bread, cellulose, sunflower oil (ratio unspecified)	46.0	-	32.0	-	-	-
Bava [21]	RS 1: Okara	51.2	-	-	-	-	-
	RS 2: Maize distillers	53.4	-	-	-	-	-
	RS 3: Brewers’ grains (uncharacterised)	54.1	-	-	-	-	-
Chia [22]	RS 1: Spent barley	37.4	-	33.2	-	-	-
	RS 2: Spent barley, brewer’s yeast	41.9	-	22.5	-	-	-
	RS 3: Spent barley, brewer’s yeast, molasses	31.7	-	49.0	-	-	-
	RS 4: Spent malted barley	39.9	-	21.1	-	-	-
	RS 5: Spent malted barley, brewer’s yeast	41.3	-	17.1	-	-	-
	RS 6: Spent malted barley, brewer’s yeast, molasses	29.9	-	39.3	-	-	-
	RS 7: Spent malted corn	40.6	-	25.5	-	-	-
	RS 8: Spent malted corn, brewer’s yeast	39.8	-	21.1	-	-	-
	RS 9: Spent malted corn, brewer’s yeast, molasses	31.8	-	42.3	-	-	-
	RS 10: Spent sorghum, barley	40.3	-	29.7	-	-	-
	RS 11: Spent sorghum, barley, brewer’s yeast	45.7	-	9.5	-	-	-
	RS 12: Spent sorghum, barley, brewer’s yeast, molasses	44.6	-	11.4	-	-	-
Danieli ^^^ [23]	RS 1: Control—ground corn, wheat bran, dehydrated alfalfa (5:2:3)	34.0	-	-	-	-	-
	RS 2: High non-fibre carbohydrate—ground barley, wheat bran, dehydrated alfalfa (6.8:2:1.2)	22.2	-	-	-	-	-
	RS 3: High fibre carbohydrate—ground barley, wheat middlings, dehydrated alfalfa, wheat straw (1.6:5:1:2.4)	34.7	-	-	-	-	-
	RS 4: High protein—ground barley, wheat middlings, dehydrated alfalfa (1.5:5.5:3)	34.2	-	-	-	-	-
Ewald [24]	RS 1: Bread	39.2	-	57.8	-	-	-
	RS 2: Fish *O. mykiss,* wheat (5:1)	52.6	-	46.7	-	-	-
	RS 3: Food waste (uncharacterised)	36.6	-	40.7	-	-	-
	RS 4: Fresh mussels *M. edulis*	44.6	-	33.1	-	-	-
	RS 5: Bread, fresh mussels *M. edulis* (9:1)	32.8	-	20.4	-	-	-
	RS 6: Bread, fresh mussels *M. edulis* (8:2)	34.2	-	19.6	-	-	-
	RS 7: Bread, fresh mussels *M. edulis* (7:3)	33.8	-	17.9	-	-	-
	RS 8: Bread, fresh mussels *M. edulis* (6:4)	36.1	-	17.9	-	-	-
	RS 9: Bread, fresh mussels *M. edulis* (5:5)	37.9	-	16.1	-	-	-
Gold [25]	RS 1: Mill by-products (uncharacterised)	42.1	-	-	-	-	-
	RS 2: Canteen waste—mix vegetables with/without dressing, sausage, offal (ratio unspecified)	36.1	-	-	-	-	-
	RS 3: Poultry waste	31.5	-	-	-	-	-
	RS 4: Vegetable canteen waste—mix vegetables with/without dressing	24.5	-	-	-	-	-
	RS 5: Mixed food waste—(1:1:1 of RS1:RS2:RS3)	28.6	-	-	-	-	-
Liland [26]	RS 1: Wheat	40.0	-	33.8	-	-	-
	RS 2: Wheat, brown algae *A. nodosum* (9:1)	37.9	-	-	-	-	-
	RS 3: Wheat, brown algae *A. nodosum* (8:2)	35.9	-	-	-	-	-
	RS 4: Wheat, brown algae *A. nodosum* (7:3)	35.3	-	-	-	-	-
	RS 5: Wheat, brown algae *A. nodosum* (6:4)	33.5	-	-	-	-	-
	RS 6: Wheat, brown algae *A. nodosum* (5:5)	33.7	-	22.2	-	-	-
	RS 7: Wheat, brown algae *A. nodosum* (4:6)	37.4	-	-	-	-	-
	RS 8: Wheat, brown algae *A. nodosum* (3:7)	42.3	-	-	-	-	-
	RS 9: Wheat, brown algae *A. nodosum* (2:8)	41.0	-	-	-	-	-
	RS 10: Wheat, brown algae *A. nodosum* (1:9)	39.3	-	-	-	-	-
	RS 11: Brown algae *A. nodosum*	41.3	-	8.1	-	-	-
Liu * [27]	RS 1: Brewery by-product (uncharacterised)	49.9	-	33.7	-	-	-
Lopes [28]	RS 1: Bread	28.0	-	-	-	-	-
	RS 2: Bread, fish *O. mykiss,* (9.5:0.5)	39.1	-	-	-	-	-
	RS 3: Bread, fish *O. mykiss,* (9:1)	42.7	-	-	-	-	-
	RS 4: Bread, fish *O. mykiss,* (8.5:1.5)	44.8	-	-	-	-	-
Meneguz ^^^ [29]	RS 1: Fruit and vegetable mix—celery, oranges, peppers (4.3:2.9:2.8)	41.9	-	25.3	-	-	-
	RS 2: Fruit—apples, oranges, apple leftovers, strawberries, mandarins, pears, kiwis, bananas, lemons (4.8:1.5:1.4:0.7:0.5:0.4:0.3:0.2:0.2)	30.8	-	39.8	-	-	-
	RS 3: Winery by-products—grape seeds, pulp, skins, stems, leaves (unspecified ratio)	34.4	-	28.7	-	-	-
	RS 4: Brewery by-products—barley brewers’ grains	53.0	-	28.3	-	-	-
Oonincx [30]	RS 1: High protein high fat—spent grains, beer yeast, cookie remain (6:2:2)	46.3	-	24.1	-	-	-
	RS 2: High protein low fat—beer yeast, potato steam peelings, beet molasses (5:3:2)	43.5	-	24.9	-	-	-
	RS 3: Low protein high fat—cookie remains, bread (5:5)	38.8	-	27.4	-	-	-
	RS 4: Low protein low fat—potato steam peelings, beet molasses, bread (3:2:5)	38.3	-	32.9	-	-	-
Salomone [31]	RS 1: Food waste—vegetable, meat/fish, bread/pasta/rice, other (6.5:0.5:2.5:0.5)	48.0	-	35.0	-	-	-
Shumo [32]	RS 1: Kitchen waste—potato peelings, carrot, rice, bread debris (ratio unspecified)	33.0	-	-	-	-	-
	RS 2: Brewery by-product spent grain	41.3	-	-	-	-	-
Spranghers ^^^ [33]	RS 1: Restaurant waste—potato, rice, pasta, vegetables (ratio unspecified)	43.1	-	-	-	-	-
Tinder [34]	RS 1: Sorghum	44.1	-	-	-	-	-
	RS 2: Sorghum, cowpea (7.5:2.5)	44.9	-	-	-	-	-
	RS 3: Sorghum, cowpea (5:5)	45.4	-	-	-	-	-
	RS 4: Sorghum, cowpea (2.5:7.5)	46.1	-	-	-	-	-
	RS 5: Cowpea	47.3	-	-	-	-	-
Tschirner [35]	RS 1: Carbohydrate—wheat middlings	37.2	-	-	-	-	-
	RS 2: Protein—dried distillers’ grains with soluble	44.6	-	-	-	-	-
	RS 3: Fibre—sugar beet	52.3	-	-	-	-	-
Cappellozza [36]	RS 1: Fruit and vegetable mix—zucchini, apple, potato, green beans, carrot, pepper, orange, celery, kiwi, plum, eggplant (unspecified ratio)	39.4	-	35.6	-	-	-
Jucker * [37]	RS 1: Fruit—apple, pear, orange (3.3:3.3:3.3)	-	11.7	-	21.0 ^≈^	-	-
	RS 2: Vegetable—lettuce, string green beans, cabbage (3.3:3.3:3.3)	-	13.2	-	2.0 ^≈^	-	-
	RS 3: Fruit and vegetable mix—(1:1 of RS1:RS2)	-	17.6	-	12.0 ^≈^	-	-
Lalander [38]	RS 1: Food waste (uncharacterised)	39.2	-	-	-	-	-
	RS 2: Fruit and vegetable mix-lettuce, apple, potato (5:3:2)	41.3	-	-	-	-	-
Nguyen * [39]	RS 1: Kitchen waste (animal and plant matter)	21.2	-	-	-	-	-
	RS 2: Fruit and vegetable (uncharacterised)	12.9	-	2.2	-	8.4	-
	RS 3: Fish (uncharacterised)	19.4	-	11.6	-	12.7	-
Barroso [40]	RS 1: Fish waste *S. aurita*-reared 1 day	77.4	-	-	-	-	-
	RS 2: Fish waste *S. aurita*-reared 2 days	78.8	-	-	-	-	-
	RS 3: Fish waste *S. aurita*-reared 4 days	75.4	-	-	-	-	-
	RS 4: Fish waste *S. aurita*-reared 6 days	50.6	-	-	-	-	-
	RS 5: Fish waste *S. aurita*-reared 8 days	71.3	-	-	-	-	-
	RS 6: Fish waste *S. aurita*-reared 10 days	61.5	-	-	-	-	-
	RS 7: Fish waste *S. aurita*-reared 12 days	61.8	-	-	-	-	-
Surendra [41]	RS 1: Food waste (uncharacterised)	43.7	-	31.8	-	12.3	-

Results presented as a percentage of total BSF larvae biomass unless indicated. Liland et al. [26] presented as total sum of amino acids. ^≈^ approximate figure; * indicative of original article presenting data as g/100g BSF larvae biomass, ^^^ indicative of original article presenting data as g/kg BSF larvae biomass. Dashes used to indicate where data is unreported in the original article, DM dry matter, FW fresh weight.

**Table 4 insects-12-00608-t004:** Essential Amino Acid Profile of Black Soldier Fly Larvae Reared on Food Waste.

Author	Rearing Substrate (RS) (Mixture Ratio)	Black Soldier Fly Larvae								
		Essential Amino Acids								
		Histidine (%)	Isoleucine (%)	Leucine (%)	Lysine (%)	Methionine (%)	Phenylalanine (%)	Threonine (%)	Tryptophan (%)	Valine (%)
Liland [26]	RS 1: Wheat	2.8	3.9	6.4	6.2	1.7	4.0	3.9	-	5.8
	RS 2: Wheat, brown algae *A. nodosum* (9:1)	2.6	4.0	6.6	6.2	1.6	3.9	4.0	-	6.0
	RS 3: Wheat, brown algae *A. nodosum* (8:2)	2.7	4.0	6.7	5.9	1.7	4.3	4.1	-	6.0
	RS 4: Wheat, brown algae *A. nodosum* (7:3)	2.4	3.9	6.6	6.0	1.6	3.8	4.0	-	6.0
	RS 5: Wheat, brown algae *A. nodosum* (6:4)	2.5	4.0	6.6	5.5	1.5	3.7	3.9	-	5.7
	RS 6: Wheat, brown algae *A. nodosum* (5:5)	2.4	4.0	6.7	5.6	1.4	3.4	4.0	-	5.7
	RS 7: Wheat, brown algae *A. nodosum* (4:6)	2.5	4.1	6.9	5.5	1.5	3.6	4.1	-	5.6
	RS 8: Wheat, brown algae *A. nodosum* (3:7)	2.3	3.8	6.3	5.6	1.3	3.0	3.7	-	5.4
	RS 9: Wheat, brown algae *A. nodosum* (2:8)	2.5	3.8	6.3	5.4	1.4	3.2	3.9	-	5.4
	RS 10: Wheat, brown algae *A. nodosum* (1:9)	2.5	3.7	6.2	5.5	1.3	3.0	3.9	-	5.5
	RS 11: Brown algae *A. nodosum*	2.7	3.8	6.2	5.6	1.4	3.2	3.9	-	5.5
Shumo ^+^ [32]	RS 1: Kitchen waste—potato peelings, carrot, rice, bread debris (ratio unspecified)	0.3	0.3	0.3	0.5	0.8	0.5	-	-	0.1
	RS 2: Brewery by-product spent grain	0.5	0.2	0.4	0.5	0.7	0.2	-	-	0.9
Spranghers ^^^ [33]	RS 1: Restaurant waste—potato, rice, pasta, vegetables (ratio unspecified)	1.4	1.9	3.1	2.3	0.7	1.6	1.6	0.5	2.8
Tschirner [35]	RS 1: Carbohydrate—wheat middlings	3.3	4.2	6.6	5.9	1.6	3.6	3.9	-	5.7
	RS 2: Protein—dried distillers’ grains with soluble	-	-	-	-	-	-	-	-	-
	RS 3: Fibre—sugar beet	-	-	-	-	-	-	-	-	-
Cappellozza ^+^ [36]	RS 1: Fruit and vegetable mix—zucchini, apple, potato, green beans, carrot, pepper, orange, celery, kiwi, plum, eggplant (unspecified ratio)	1.2	1.6	2.7	2.0	1.8	1.9	2.2	0.4	2.5
Lalander ^^^ [38]	RS 1: Food waste (uncharacterised)	2.9	4.1	6.8	8.3	1.8	4.0	3.9	1.4	5.8
	RS 2: Fruit and vegetable mix—lettuce, apple, potato (5:3:2)	2.6	4.3	6.7	5.1	1.5	3.5	3.5	1.4	6.0
Surendra ^ww^ [41]	RS 1: Food waste (uncharacterised)	1.7	1.5	2.3	2.2	0.9	1.5	1.5	-	2.4

Results presented as a percentage of the BSF larvae total protein content unless indicated. Liland et al. [26] presented as percentage of total sum of amino acids. ^^^ indicative of original article presenting data as g/kg of the BSF larvae total protein content, ^+^ indicative of original article presenting data as mg/g of the total BSF larvae total protein. Dashes used to indicate where data is unreported in the original article. ^WW^ wet weight.

**Table 5 insects-12-00608-t005:** Non-Essential Amino Acid Profile of Black Soldier Fly Larvae Reared on Food Waste.

Author	Rearing Substrate (RS) (Mixture Ratio)	Black Soldier Fly Larvae										
		Non-Essential Amino Acids										
		Alanine (%)	Arginine (%)	Aspartate (%)	Cysteine (%)	Glutamate (%)	Glutamic acid (%)	Glutamine (%)	Glycine (%)	Proline (%)	Serine (%)	Tyrosine (%)
Liland [26]	RS 1: Wheat	6.2	4.6	9.4	-	-	10.3	-	4.6	5.3	4.0	5.7
	RS 2: Wheat, brown algae *A. nodosum* (9:1)	6.6	4.5	9.2	-	-	11.8	-	5.0	5.8	4.3	5.6
	RS 3: Wheat, brown algae *A. nodosum* (8:2)	6.5	4.9	8.6	-	-	11.3	-	5.2	5.8	4.4	5.6
	RS 4: Wheat, brown algae *A. nodosum* (7:3)	6.6	4.5	8.5	-	-	11.9	-	5.1	5.8	4.4	5.1
	RS 5: Wheat, brown algae *A. nodosum* (6:4)	6.5	5.0	8.1	-	-	11.8	-	5.0	5.9	4.5	4.9
	RS 6: Wheat, brown algae *A. nodosum* (5:5)	6.8	4.6	8.3	-	-	12.3	-	4.8	5.9	4.6	4.3
	RS 7: Wheat, brown algae *A. nodosum* (4:6)	6.9	5.3	8.2	-	-	12.8	-	5.3	6.1	4.9	4.4
	RS 8: Wheat, brown algae *A. nodosum* (3:7)	6.6	4.5	7.9	-	-	11.7	-	4.7	5.4	4.3	3.6
	RS 9: Wheat, brown algae *A. nodosum* (2:8)	6.5	5.0	8.0	-	-	11.9	-	5.0	5.5	4.4	4.1
	RS 10: Wheat, brown algae *A. nodosum* (1:9)	6.3	4.7	8.0	-	-	11.6	-	5.0	5.0	4.4	4.0
	RS 11: Brown algae *A. nodosum*	6.4	6.5	8.3	-	-	11.9	-	4.3	5.2	4.3	4.2
Shumo ^+^ [32]	RS 1: Kitchen waste—potato peelings, carrot, rice, bread debris (ratio unspecified)	-	0.5	-	-	-	0.6	0.8	-	0.5	-	0.5
	RS 2: Brewery by-product spent grain	-	0.3	-	-	-	0.3	0.0	-	0.2	-	0.3
Spranghers ^^^ [33]	RS 1: Restaurant waste—potato, rice, pasta, vegetable (ratio unspecified)	2.8	2.0	3.7	0.2	-	4.6	-	2.5	2.5	1.6	-
Tschirner [35]	RS 1: Carbohydrate—wheat middlings	7.8	4.8	8.2	0.9	-	11.8	-	5.6	6.2	4.3	5.1
	RS 2: Protein—dried distillers’ grains with soluble	-	-	-	-	-	-	-	-	-	-	-
	RS 3: Fibre—sugar beet	-	-	-	-	-	-	-	-	-	-	-
Cappellozza ^+^ [36]	RS 1: Fruit and vegetable mix—zucchini, apple, potato, green beans, carrot, pepper, orange, celery, kiwi, plum, eggplant (unspecified ratio)	3.9	4.8	3.8	0.5	-	6.5	-	2.3	1.5	1.8	2.0
Lalander ^^^ [38]	RS 1: Food waste (uncharacterised)	5.9	4.9	9.1	0.5	9.8	-	-	5.3	5.1	4.1	6.0
	RS 2: Fruit and vegetable mix—lettuce, apple, potato (5:3:2)	5.5	4.5	8.1	0.5	9.5	-	-	5.2	5.3	3.9	5.5
Surendra ^ww^ [41]	RS 1: Food waste (uncharacterised)	2.7	2.2	-	1.1	-	-	-	2.5	2.1	1.5	2.4

Results presented as a percentage of the BSF larvae total protein content unless indicated. Liland et al. [26] presented as percentage of total sum of amino acids. ^^^ indicative of original article presenting data as g/kg of the BSF larvae total protein content, ^+^ indicative of original article presenting data as mg/g of the total BSF larvae total protein. Dashes used to indicate where data is unreported in the original article. ^WW^ wet weight.

## Data Availability

The data presented in this study are available in The Influence of Food Waste Rearing Substrates on Black Soldier Fly Larvae Protein Composition: A Systematic Review.

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
