# Peer review of "The Influence of Food Waste Rearing Substrates on Black Soldier Fly Larvae Protein Composition: A Systematic Review"

_insects, 2021, doi:10.3390/insects12070608_

Round 1

Reviewer 1 Report

Line 19: move to the conclusion section.

According to the MDPI publishing guidelines, Abstract greatly differs from Simple Summary. It is not advisable to repeat the same sentences in both sections.

Line 72:  fish feed supplements.

Aquaculture, 2020, 522, 735136

Fish and Shellfish Immunology, 2021, 111, pp. 111–118

Line 95: why the authors collect the data from January 1st, 2000 and October 30th, 2020? 

All highlighted Latin names must be italic (see PDF file)

Do not repeat the same words throughout the whole manuscript (see PDF)

Please see the attached PDF file.

Reviewer 2 Report

This manuscript by Hopkins et al. is a systematic review of the literature about the use of food waste to rear black soldier fly (BSF) larvae. The authors set up a literature search procedure that allowed them to select 23 articles among 1712 records identified in current databases and published between 2000 and 2020. These papers were then analyzed in detail and the results are herein reported.

The article is well written and provides an original and comprehensive view of all the main studies that have explored the use of various kinds of organic substrates to grow BSF larvae so far. It contains a huge amount of data that can be readily compared and used by those who aim at exploiting BSF for the conversion of organic waste. The report is very detailed and accurate and various parameters related to the feeding substrate and the insect have been considered, including abiotic factors as temperature, RH, and light. Finally, shortcomings and critical aspects that, on one side could affect compositional analyses of BSF (such as analytic methods used for protein content evaluation and larval processing procedures) and, on the other side, could help to better exploit this insect for CORS (lack of studies that correlate the composition of the feeding substrate with amino acid content of the insect) are discussed.

I have just few suggestions to improve the review:

1) Simple summary and Abstract are almost identical. They should be rewritten.

2) Line 10. I suggest to change the term “to form” with “to generate” or “to obtain”.

3) Lines 544-547. While food substrate is the main factor that affects the protein/lipid composition of BSF larvae, the protein content of the insect is generally stable along the larval phase. Conversely, the protein content significantly changes at different developmental stages, as prepupal and pupal phase. For this reason, I suggest to smooth down this hypothesis. Instead, the authors should discuss the correlation between the composition of the larvae and the collection stage reported in different studies (larvae VS prepupae) since this could be a key aspect in the interpretation of data.   

4) Another factor that can affect the feeding activity of BSF larvae is their rearing density. This information should be added to one of the tables, if known.

5) Lines 621-624. This sentence is not clear to me since the protein content of the larvae significantly changed in studies that used different feeding substrates.

6) Supplementary Table 3 should be revised (see misalignment in references 36 and 37).

Reviewer 3 Report

Dear Authors, 

I reviewed your manuscipt "The influence of food waste rearing substrates on Black Soldier Fly larvae protein composition: a systematic review".  I think you should review the MS  before resubmitting it. In fact, numerous parameters relating to BSF's growth are taken into consideration and deserve an extra comment. For example, it would be useful to find a correlation between the different parameters and the final protein composition. Also comments and discussion on the protein content of the substrates and the final content of the larvae are missing.

You can find my suggestion in the pdf file.

Round 2

Reviewer 3 Report

Dear Authors, you can find final suggestions in the attach file.
